# On the Inductive Bias of Neural Networks for Learning Read-once DNFs

**Ido Bronstein**[1]  **Alon Brutzkus**[1]  **Amir Globerson**[1]

[1]Blavatnik School of Computer Science , Tel Aviv University, Israel

## Abstract

Learning functions over Boolean variables is a fundamental problem in machine learning. But not much is known about learning such functions using neural networks. Here we focus on learning read-once disjunctive normal forms (DNFs) under the uniform distribution with a convex neural network and gradient methods. We first observe empirically that gradient methods converge to compact solutions with neurons that are aligned with the terms of the DNF. This is despite the fact that there are many zero training error networks that do not have this property. Thus, the learning process has a clear inductive bias towards such logical formulas. Following recent results which connect the inductive bias of gradient flow (GF) to Karush-Kuhn-Tucker (KKT) points of minimum norm problems, we study these KKT points in our setting. We prove that zero training error solutions that memorize training points are not KKT points and therefore GF cannot converge to them. On the other hand, we prove that globally optimal KKT points correspond exactly to networks that are aligned with the DNF terms. These results suggest a strong connection between the inductive bias of GF and solutions that align with the DNF. We conclude with extensive experiments which verify our findings.

## 1 INTRODUCTION

The training objective of overparameterized neural networks is non-convex and contains multiple global minima with different generalization properties. Therefore, just minimizing the training objective does not guarantee good generalization performance. Nonetheless, neural networks trained in practice with gradient-based methods show good test performance across numerous tasks [Krizhevsky et al., 2012, Silver et al., 2016], suggesting an *inductive bias* towards desirable solutions. Understanding this inductive bias and how it depends on the algorithm, architecture and data is one of the major open problems in machine learning [Zhang et al., 2017, Neyshabur et al., 2018].

In recent years, there have been major efforts to tackle this challenge. One line of works considers the Neural Tangent Kernel (NTK) approximation of neural networks which reduces to a convex optimization problem [Jacot et al., 2018]. However, it has been shown that the NTK approximation is limited and does not accurately model neural networks as they are used in practice [Yehudai and Shamir, 2019, Daniely and Malach, 2020].

Other works tackle the non-convexity directly, usually in very simplified settings [e.g., diagonal linear networks Woodworth et al., 2019] or for special cases such as regression with 2-layer models and Gaussian distributions [Li et al., 2020] or infinitely wide two-layer networks [Chizat and Bach, 2020].

Here we focus on the important problem of learning Boolean functions with neural networks. While much is known about this problem from a computational and statistical perspective, little is understood on how they can be learned with neural networks, and in particular on the inductive bias of gradient descent in this case. In computational learning theory, the problem of learning disjunctive normal forms (DNFs) has a long history. Learning DNFs is hard [Pitt and Valiant, 1988] and the best known algorithms for learning DNFs under the uniform distribution run in quasi-polynomial time [Verbeurgt, 1990]. On the other hand, for learning *read-once* DNFs under the uniform distribution there exist efficient learning algorithms [Mansour and Schain, 2001].[1] Therefore, it is interesting to understand whether neural networks can learn read-once DNFs under the uniform distribution and this motivates the study of the inductive bias in this case.

---

[1]In a read-once DNF each literal appears at most once. See Section 3 for a formal definition.

*Accepted for the 38th Conference on Uncertainty in Artificial Intelligence* (UAI 2022).

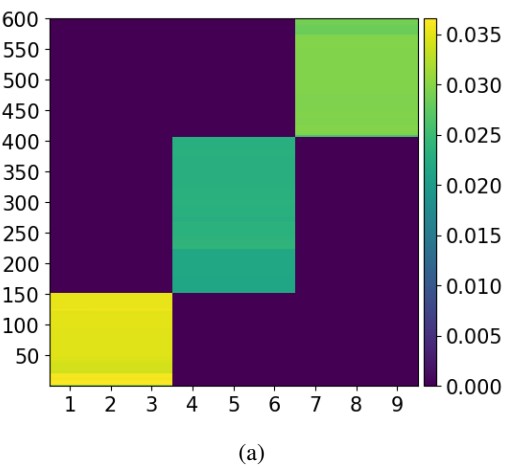
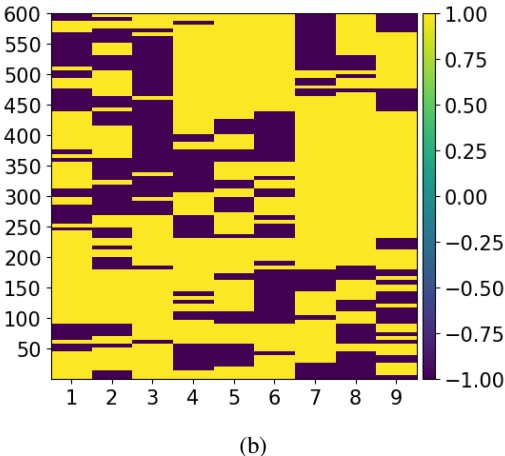

|       (a)       |       (b)       |

Figure 1: The weight vectors for two convex networks that perfectly fit 250 training samples labeled with the read-once DNF: $(x_1 \wedge x_2 \wedge x_3) \vee (x_4 \wedge x_5 \wedge x_6) \vee (x_7 \wedge x_8 \wedge x_9)$. (a) A network trained with SGD using small Gaussian initialization. The weights can be seen to be well aligned with the DNF terms, and the test accuracy is 100%. (b) A network whose weight vectors are equal to the positive samples in the training set. This network "memorizes" the training data and fits it perfectly. However, the test error is 81.6%.

We focus specifically on a simple neural architecture with one-hidden layer and ReLU activations, and output weights equal to one. We refer to it as a convex network, because the network output is a convex function of its inputs in this case. It is easy to see that this architecture is sufficiently expressive for learning DNFs. We show that for learning read-once DNFs there exist solutions that perfectly classify the training set with significantly different properties. Specifically, solutions which memorize the training points in their neurons, and other solutions whose neurons align exactly with the terms of the DNF, which we call DNF recovery solutions. Figure 1a-b shows an example of these solutions.[2]

Our first empirical finding is that SGD with small initialization converges to a DNF-recovery solution. This indicates a strong inductive bias of gradient methods towards simple logical forms in this case. We further observe that this bias allows the convex network to generalize better than algorithms designed specifically for learning read-once DNFs. Together these empirical observations establish neural nets as an attractive approach to learning read-once DNFs.

Given the above, we ask what can explain this inductive bias of gradient methods, and what theoretical guarantees can be obtained for its performance. We turn to recent line of works [Soudry et al., 2018, Ji and Telgarsky, 2019a, Lyu and Li, 2020, Ji and Telgarsky, 2020, Chizat and Bach, 2020] which study the inductive bias of gradient flow (GF) in several settings. Their results suggest that the inductive bias of GF is to Karush-Kuhn-Tucker (KKT) points or global solutions of

minimum norm problems (or analogously maximum margin problems). Motivated by these results, we prove that any norm minimizing solution in our setting is a DNF Recovery solution, strongly suggesting why it is that GF converges to it. We further strengthen this result by proving that memorizing solutions (namely, solutions where there are neurons that are only activated by specific inputs, as in Figure 1b) are not KKT points of the min-norm problem. Therefore, GF will not converge to these.

We corroborate our findings with empirical results which show that our conclusions hold more broadly. Specifically, we perform experiments on DNFs of higher dimension and standard one-hidden layer neural networks. Taken together, our results demonstrate that gradient methods can recover simple descriptions of Boolean functions from data, which results in good generalization performance and may have important implications on the question of interpretability.

## 2  RELATED WORK

Recently, several works studied the inductive bias of neural networks and showed connections between gradient methods and margin maximization [Chizat and Bach, 2020, Lyu and Li, 2020, Nacson et al., 2019, Ji and Telgarsky, 2020]. These works motivate our theoretical analysis of minimum norm solutions. In our theoretical analysis, we apply the results of Lyu and Li [2020], Ji and Telgarsky [2020], which show that GF is biased towards KKT points of min-norm problems.

Other works study fully connected neural networks under certain assumptions on the data such as linearly separable data [Brutzkus et al., 2018, Sarussi et al., 2021, Frei et al.,

---

[2] The position $(j, i)$ in the figure represents the value of entry $j$ of the weight vector $w_i$ (see Eq. (1)) and the color represents its value at the end of the learning process.

2021] or Gaussian data [Safran and Shamir, 2018, Du et al., 2019]. Malach and Shalev-Shwartz [2020] show that certain structured Boolean circuits can be learned with a network architecture that is specialized for their data structure.

Fully connected networks were also analyzed via the NTK approximation [Jacot et al., 2018, Du et al., 2019, 2018, Arora et al., 2019, Ji and Telgarsky, 2019b, Cao and Gu, 2019, Jacot et al., 2018, Fiat et al., 2019, Allen-Zhu et al., 2019, Li and Liang, 2018, Daniely et al., 2016]. However, Yehudai and Shamir [2019], Daniely and Malach [2020] have highlighted limitations of the NTK framework, suggesting that it does not accurately model neural networks as they are used in practice.

Another line of works [Saad and Solla, 1996, Goldt et al., 2019, Tian, 2019] studies neural networks in student-teacher regression settings and shows a "specialization" effect, where a subset of student neurons aligns with teacher neurons. The main difference from our setting is that we consider classification on binary data, and they consider regression tasks on non-discrete data (e.g., Gaussian). We note that classification settings present unique theoretical challenges for studying inductive bias of gradient methods [Montanari et al., 2019].

In a recent result, Phuong and Lampert [2021] provide an end-to-end optimization analysis of a two-layer ReLU network on orthogonally separable data (which is a simplified setup of linearly separable data). They consider the cross-entropy loss and their analysis implies that neurons specialize to certain directions. We focus on a significantly more challenging setting, where the training data corresponds to a read-once DNF, and is generally not linearly separable.

An inductive bias towards specializing solutions has also been observed in Brutzkus and Globerson [2019] and proved for a simple setup with nonlinear data and a convolutional neural network. The notion of specialization is also related to the notion of "collapse" [Papyan et al., 2020]. We note that in our setting we do not observe the collapse phenomenon since the hidden-layer representations of the positive samples are not all in one small cluster (e.g., see Figure 1a).[3]

Rudin [2019] argues that methods for explaining large neural networks should be avoided because networks are too complex for humans to understand. However, we show, albeit in a restricted setting, that learned networks can be rather simple, and are easily mapped to the underlying DNF.

---

[3]For the model in the figure, the representations of the positive points create 8 different clusters which correspond to each possible combination of the three terms.

## 3  PROBLEM FORMULATION

**DNFs and Read-Once DNFs:**  In what follows, we use $[n]$ to denote the set $\{1, 2, ..., n\}$. Let $\mathcal{X} = \{\pm 1\}^D$, where $D$ is the number of variables, and let $\mathcal{Y} = \{\pm 1\}$. Boolean functions [e.g., see O'Donnell, 2014] are usually defined on inputs with entries in $\{0, 1\}$ to an output in $\{0, 1\}$. In this work, we consider DNFs on inputs with entries in $\{\pm 1\}$ and output in $\{\pm 1\}$.

A DNF is a disjunction of conjunctions over one or more literals. See the DNF in Figure 1 for an example. For convenience, we will use the following notation for DNFs: A DNF with $K$ terms will be defined via $K$ indicator vectors $\boldsymbol{t}_1^*, ..., \boldsymbol{t}_K^* \in \{0, 1\}^D$. We refer to each $\boldsymbol{t}_n^*$ as a *term* and define its set of active indices by $\mathbb{A}_n = \{j \in [D] \mid t_{nj}^* = 1\}$, where $t_{nj}^*$ is the $j$th entry of $\boldsymbol{t}_n^*$. The corresponding DNF will be given by the function $f^* : \mathcal{X} \to \mathcal{Y}$ as follows: $f^*(\boldsymbol{x}) = 1$ if $\exists n \in [K]$ $s.t.$ $\boldsymbol{x} \cdot \boldsymbol{t}_n^* = |\mathbb{A}_n|$, and otherwise $f^*(\boldsymbol{x}) = -1$. Notice that $f^*$ is monotone. We say that a sample $\boldsymbol{x} \in \mathcal{X}$ satisfies the term $\boldsymbol{t}_n^*$ if $\boldsymbol{x} \cdot \boldsymbol{t}_n^* = |\mathbb{A}_n|$. We refer to $|\mathbb{A}_n|$ as the size of the term $\boldsymbol{t}_n^*$.

To compare our notation with the standard one, for example, the DNF $(x_1 \wedge x_2) \vee (x_3 \wedge x_4)$ with 4 inputs has terms $\boldsymbol{t}_1^* = (1, 1, 0, 0)$ and $\boldsymbol{t}_2^* = (0, 0, 1, 1)$. We will use the standard notation when convenient (e.g., as in Figure 1).

In this work we will focus on ***read-once*** DNFs where for all $i \neq j \in [K]$, $\mathbb{A}_i \cap \mathbb{A}_j = \emptyset$ and the sizes of all the terms are greater than 1.

**Learning Setup:**  Let $\mathcal{D}$ be a distribution on $\mathcal{X} \times \mathcal{Y}$. We assume that for $(\boldsymbol{x}, y) \sim \mathcal{D}$, $\boldsymbol{x}$ is sampled uniformly over the hypercube $\{\pm 1\}^D$ and $y = f^*(\boldsymbol{x})$, where $f^*$ is a monotone read-once DNF. [4]

We consider learning $f^*$ given a training set $\mathbb{S} \subseteq \mathcal{X} \times \mathcal{Y}$, where for each $(\boldsymbol{x}, y) \in \mathbb{S}$, $\boldsymbol{x}$ is sampled IID from $\mathcal{D}$ and $y = f^*(\boldsymbol{x})$. Denote $\mathbb{S}_x = \{\boldsymbol{x} \mid (\boldsymbol{x}, y) \in \mathbb{S}\}$, the positive samples by $\mathbb{S}_+ = \{\boldsymbol{x} \mid (\boldsymbol{x}, 1) \in \mathbb{S}\}$, the negative samples by $\mathbb{S}_- = \{\boldsymbol{x} \mid (\boldsymbol{x}, -1) \in \mathbb{S}\}$ and the number of samples by $m = |\mathbb{S}|$. In some cases we will consider the population case where $\mathbb{S}_x = \mathcal{X}$.

**Neural Architecture:**  We consider a **convex** one-hidden layer neural network (NN) with $r$ hidden units and parameters $\boldsymbol{\theta} = (\boldsymbol{W}, \boldsymbol{b}, c) \in \mathbb{R}^{rD} \times \mathbb{R}^r \times \mathbb{R}$ which is defined by:

$$N(\boldsymbol{x}; \boldsymbol{\theta}) = \sum_{i \in [r]} \sigma(\boldsymbol{w}_i \cdot \boldsymbol{x} + b_i) + c \qquad (1)$$

---

[4]In the case of the uniform distribution and read-once DNFs, we can assume monotone DNFs WLOG. This follows since any negated literal can be replaced with the original literal (without negation) and all our results still hold. Note that for non-read-once DNFs, this will not work because a variable can appear both positively and negatively and flipping its value will not make the DNF monotone.

where $\sigma(x) = max\{0, x\}$ is the ReLU function, $\boldsymbol{w}_i$ is the $i^{\text{th}}$ row of $\boldsymbol{W}$ and $b_i$ is the $i^{\text{th}}$ entry of $\boldsymbol{b}$. We also use a scalar trainable bias $c \in \mathbb{R}$ in the second layer to allow for negative outputs.

The resulting network is positive homogeneous, and thus recent results on such networks can be applied [Lyu and Li, 2020, Ji and Telgarsky, 2020]. Note that the network is a convex function of its weights because it is a sum of convex ReLU functions [Amos et al., 2017].

**Loss Minimization:** To learn $f^*$ we consider minimizing the following loss:

$$L(\boldsymbol{\theta}) = \frac{1}{m} \sum_{(\boldsymbol{x}, y) \in \mathbb{S}} \ell\left(yN(\boldsymbol{x}; \boldsymbol{\theta})\right) \qquad (2)$$

where $\ell(z) = \log\left(1 + e^{-z}\right)$ is the binary cross entropy loss. We note that $L(\boldsymbol{\theta})$ is generally non-convex (even though the network $N$ is convex). For our theoretical analyses we consider Gradient Flow (GF). We denote the initialization of GF by $\boldsymbol{\theta}^{(0)} = \left(\boldsymbol{W}^{(0)}, \boldsymbol{b}^{(0)}, c^{(0)}\right)$ and the weights at iteration $t$ by $\boldsymbol{\theta}^{(t)} = \left(\boldsymbol{W}^{(t)}, \boldsymbol{b}^{(t)}, c^{(t)}\right)$. If the iteration index is clear from context we omit it and use $\boldsymbol{\theta} = (\boldsymbol{W}, \boldsymbol{b}, c)$.

Recall that gradient flow is the infinitesimal step limit of gradient descent where $\boldsymbol{\theta}^{(t)}$ changes continuously in time and satisfies the differential inclusion $\frac{d\boldsymbol{\theta}^{(t)}}{dt} \in -\partial^\circ L(\boldsymbol{\theta}^{(t)})$ for a.e. $t$. Here $\partial^\circ L\left(\boldsymbol{\theta}^{(t)}\right)$ is the Clarke's sub-differential which is a generalization of the differential for non-differentiable functions:

$$\partial^\circ f(\boldsymbol{x}) = \text{conv}\left\{ \lim_{k \to \infty} \nabla f(\boldsymbol{x}_k) \mid \boldsymbol{x}_k \to \boldsymbol{x} \text{ and} \qquad (3) \right.$$
$$\left. f \text{ is differentiable at } \boldsymbol{x}_k \right\}$$

The differential inclusion allows to take any vector in $-\partial^\circ L(\boldsymbol{\theta}^{(t)})$ in each step of gradient flow. In our case, the differential inclusion has multiple possible values when the ReLU is 0 since ReLU has multiple sub-gradients at 0. For our theoretical results, we will assume that the subgradient of ReLU at 0 is determined in advance to a value $a \in [0, 1]$. This value of the subgradient is used for all neurons and in all iterations. Usually $a$ is set to be either 0 or 1. This assumption corresponds to the common way gradient descent runs in practice. We provide a formal definition of this assumption in the supplementary.

Next, we define solutions which perfectly classify the training set. We will consider solutions with a margin constraint, since this will be convenient when we discuss minimum norm solutions.

**Definition 3.1.** *We say that a solution $\boldsymbol{\theta}$ is perfect if for all* $(\boldsymbol{x}, y) \in \mathbb{S}$, $yN(\boldsymbol{x}; \boldsymbol{\theta}) \geq 1$.

**Norm Minimization:** Multiple recent works have highlighted interesting connections between gradient methods

and norm minimization or margin maximization [Lyu and Li, 2020, Neyshabur et al., 2018, Nacson et al., 2019, Ji and Telgarsky, 2020, Chizat and Bach, 2020]. The norm minimization problem is to minimize the norm of the model weights subject to the correct classification with a margin (additional background can be found at [Boyd and Vandenberghe, 2004]). Namely, the problem is:

$$\begin{aligned} \min \quad & \sum_{i \in [r]} ||(\boldsymbol{w}_i, b_i)||_2^2 + c^2 \\ \text{s.t.} \quad & yN(\boldsymbol{x}; \boldsymbol{\theta}) \geq 1 \; , \; \forall (\boldsymbol{x}, y) \in \mathbb{S} \end{aligned} \qquad (4)$$

It was shown [Lyu and Li, 2020, Ji and Telgarsky, 2020] that under certain conditions, gradient flow converges to KKT points of the optimization problem in Eq. (4).

## 3.1 EXPRESSIVE POWER

Here we show that the network in Eq. (1) has the expressive power to implement *any* Boolean function over $\mathcal{X}$. Therefore, in terms of expressive power, the network is suitable for learning Boolean functions and has the same expressive power for implementing Boolean functions as a standard one-hidden layer NN.

**Theorem 3.1.** *Let* $f : \mathcal{X} \to \mathcal{Y}$. *Then, there exists* $\boldsymbol{\theta}$ *and a network $N$ in Eq. (1) with $r \leq 2^D$ neurons such that for all* $\boldsymbol{x} \in \mathcal{X}$, $\text{sign}\left(N(\boldsymbol{x}; \boldsymbol{\theta})\right) = f(\boldsymbol{x})$.

*Proof.* Let $\mathcal{X}_+ = \{\boldsymbol{x} \mid f(x) = 1\}$. Define $r = |\mathcal{X}_+|$. Then, $\mathcal{X}_+ = \{\boldsymbol{x}_1, \ldots, \boldsymbol{x}_r\}$. Define $c = -1$ and for each $i \in [r]$ define $\boldsymbol{w}_i = \boldsymbol{x}_i$ and $b_i = -D + 2$. Then $\forall \boldsymbol{x}_i \in \mathcal{X}_+$ it holds that $\sigma(\boldsymbol{w}_i \cdot \boldsymbol{x}_i + b_i) = 2$ and $\forall \boldsymbol{x} \neq \boldsymbol{x}_i$ it holds that $\sigma(\boldsymbol{w}_i \cdot \boldsymbol{x} + b_i) = 0$. Therefore $\forall \boldsymbol{x} \in \mathcal{X}_+$ we have $N(\boldsymbol{x}; \boldsymbol{\theta}) = 1$ and for $\boldsymbol{x} \in \mathcal{X} \backslash \mathcal{X}_+$ it holds that $N(\boldsymbol{x}; \boldsymbol{\theta}) = -1$, from which the claim follows. $\qquad \square$

## 3.2 EMPIRICAL PERFORMANCE

Thus far we described the setting where the ground truth function is a read-once DNF that is learned by a convex neural net. We have seen in Theorem 3.1 that the convex network is sufficiently expressive. However, this does not imply that the network can learn read-once DNFs in practice. To examine this, we performed experiments for learning read-once DNFs under the uniform distribution with the convex network. We compared its test performance to a standard two-layer neural network, and an algorithm based on Statistical Queries (SQ) for learning read-once DNFs that has polynomial sample complexity guarantees [Mansour and Schain, 2001]. We note that the convex network was implemented with a relatively small initialization. In Section 7 and in the Supplementary we conduct experiments with a convex network with large initialization which is analogous to training in the NTK regime [Chizat et al., 2019].

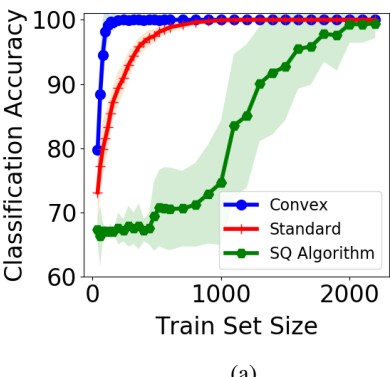
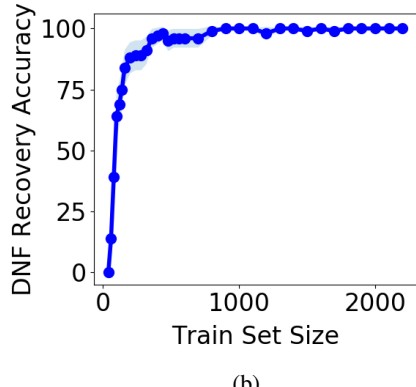

|       (a)       |       (b)       |

Figure 2: Learning the read-once DNF: $(x_1 \wedge x_2 \wedge x_3) \vee (x_4 \wedge x_5 \wedge x_6) \vee (x_7 \wedge x_8 \wedge x_9)$ (a) Test accuracy for the following models: convex neural network, standard two layer neural network and an algorithm based on statistical queries (SQ). (b) Accuracy of the DNF recovery procedure for finding exactly the true DNF from the network weights.

Figure 2a shows the evaluation results. It can be seen that the convex network outperforms the other algorithms across all training set sizes.

Therefore, together with Theorem 3.1, we conclude that the convex network we consider is a good test-bed for analyzing the inductive bias of neural networks in the setting of read-once DNFs and uniform distribution.

# 4 EMPIRICAL OBSERVATIONS ON THE INDUCTIVE BIAS

Overparameterized networks (i.e., large $r$), can fit the training data with multiple different solutions. Some have good generalization performance while others overfit. When training is performed with gradient methods, only certain solutions will be found and not others. In other words, gradient methods have an *inductive bias* towards certain solutions. Understanding this inductive bias is key for understanding the generalization performance of gradient methods in practice.

Figure 1a shows an example of the weight vectors learned by an overparameterized convex network (Eq. (1)) optimized with SGD. It can be seen that although the network has many weight vectors, they form tight clusters around the terms of the true underlying DNF. This network has perfect accuracy on both the training set and the test set.

We further devise a simple procedure to recover the DNF terms from the neurons. The procedure removes low norm neurons and rounds the weights, the exact details are provided in the supplementary. Figure 2b shows that the DNF recovery procedure can accurately find the DNF terms. In the supplementary we provide further details on this experiment and show many more examples of the inductive bias of SGD towards solutions which align with terms.

Another solution that minimizes the training error is shown in Figure 1b. In this solution, the network memorizes in its neurons the positive training points.[5] However, this network does not generalize well, and has 81% test accuracy. Thus, we see that SGD converges to the true DNF and not to a "memorization" solution, despite the latter also minimizing the training loss.

These observations raise the following intriguing questions:

1. Why do gradient methods have an inductive bias towards solutions that align with the terms of the DNF?

2. Why do gradient methods not converge to solutions that overfit and memorize training points in their neurons?

In the next sections we provide theoretical results which address these questions.

# 5 GRADIENT FLOW DOES NOT MEMORIZE

In this section, we prove that gradient flow (GF) does not converge to solutions that memorize training points. First we formally define a memorizing neuron and a memorization solution.

**Definition 5.1.** *A neuron $i \in [r]$ is a* memorizing neuron, *if there exists a sample $\hat{x} \in \mathbb{S}_x$ such that:*

$$\boldsymbol{w}_i \cdot \hat{\boldsymbol{x}} + b_i > 0 \text{ and } \forall \boldsymbol{x} \in \mathcal{X} \setminus \{\hat{\boldsymbol{x}}\} \ \boldsymbol{w}_i \cdot \boldsymbol{x} + b_i \leq 0 \quad (5)$$

*In this case, we say that neuron $i$ memorizes $\hat{x}$.*

Thus, a neuron $i$ memorizes $\hat{x}$ if it is the only point in $\mathcal{X}$ that has a positive dot product with the neuron. Since the nonlinear activation of the network is ReLU, this implies that only the point $\hat{x}$ activates neuron $i$.

---

[5]See a formal definition of memorization in the next section.

**Definition 5.2.** *$\theta$ is a memorization solution if $\theta$ is perfect (Definition 3.1) and there exists $i \in [r]$ and a sample $\hat{x} \in \mathbb{S}_x$ such that neuron $i$ memorizes $\hat{x}$.*

Note that the solution in Figure 1b is a memorization solution where all positive points in $\mathbb{S}_+$ are memorized. Definition 5.2 defines a broader set of solutions in which at least one point is memorized.

We now state the assumptions that are required for our main result. We apply recent results of Lyu and Li [2020], Ji and Telgarsky [2020], which assume that GF is in the late phase of training. Therefore, we will need the following assumption.

**Assumption 5.1.** *There exists $t_0$ s.t. $L\left(\theta^{(t_0)}\right) < \frac{\ln 2}{n}$.*

The next theorem shows that GF cannot converge to memorization solutions.

**Theorem 5.1.** *Assume that Assumption 5.1 holds and $D > 2$, $K \geq 2$. Let $\theta$ be a memorization solution. Then GF does not converge to $\theta$.*

The proof ideas is as follows. Lyu and Li [2020] and Ji and Telgarsky [2020] show that under Assumption 5.1, GF converges to a KKT point of Eq. (4). KKT points must satisfy the KKT conditions: stationarity and complementary slackness (see supplementary for details). We use this fact together with the structure of the subgradient updates to show that memorization solutions cannot satisfy the KKT conditions.

To show this, we first characterize the memorizing neuron using the following lemma:

**Lemma 5.1.** *For $D > 2$. Let $\theta$ be a solution with a neuron $i \in [r]$ that memorizes a sample $\hat{x} \in \mathbb{S}_x$, then $\theta$ satisfies the following properties:*

1. *$\hat{x}_j = \text{sign}(w_{ij})$ for all $1 \leq j \leq D$.*
2. *For $x \in \mathcal{X}$ if $w_i \cdot x + b_i = 0$ then $x \cdot \hat{x} = D - 2$.*
3. *$b_i < 0$*

Then, by complementary slackness and the non-negativity of the slack variables we show that the stationarity conditions of the weights and biases cannot hold. Thus, memorization solutions are not KKT points and GF cannot converge to them.

## 6 DNF RECOVERY AS NORM MINIMIZATION

In the previous section we have shown that GF does not converge to memorization solutions. This result followed since GF converges to KKT points of the minimum norm problem and memorization solutions are not KKT. However, we would like to understand what are the KKT points that GF does converge to.

To address this question, we focus on characterizing KKT points which are also *global minimizers* of the minimum norm problem. The reason is that there is growing theoretical evidence that GF is biased towards solutions that minimize norms (or analogously, solutions that maximize margins). This has been shown for logistic regression [Soudry et al., 2018] and linear networks [Ji and Telgarsky, 2019a].

In a setting which is closest to ours, Chizat and Bach [2020] show that GF trained on two-layer nonlinear networks converges to maximum margin solutions. Their result holds for infinite 2-homogeneous networks with squared ReLU activations. Therefore, it cannot be applied in our setting. Nonetheless, all of the aforementioned results provide a strong motivation to study minimum norm solutions in our setting, to better understand the inductive bias of GF.

Analyzing the global solutions of the optimization problem in Eq. (4) is a major challenge since the problem is non-convex. To make headway, in this section we analyze these solutions under two technical assumptions (see Assumption 6.1 and Assumption 6.2). We define DNF recovery solutions (see Definition 6.2) as solutions which are aligned with the terms of the DNF (similar to Figure 1a). We then prove our main result: that a network that globally optimizes Eq. (4) is a DNF recovery solution. This means that if GF globally optimizes Eq. (4) then it must converge to a DNF recovery solution. Furthermore, this result is in line with our experiments in Section 7 which show that GF converges to DNF recovery solutions.

We next formally define a DNF recovery solutions. We first define alignment of a neuron with a term.

**Definition 6.1.** *A neuron $i \in [r]$ is an aligning neuron with respect to a DNF $f^*$, if there exists $n \in [K]$ and $\lambda_i > 0$ such that $w_i = \lambda_i t_n^*$, and $b_i = \lambda_i(2 - \|t_n^*\|_1)$. We refer to the neuron $i$ as aligning with the term $n$, and to $\lambda_i$ as the alignment coefficient of $i$.*

Next we define a DNF recovery solution.

**Definition 6.2.** *$\theta$ is a DNF-recovery solution if $\forall n \in [K]$ there exists a set of neurons $\mathbb{I}$ such that every $i \in \mathbb{I}$ aligns with term $n$, $\sum_{i \in \mathbb{I}} \lambda_i = 1$, $\forall i_1, i_2 \in \mathbb{I}$ $\lambda_{i_1} = \lambda_{i_2}$ and all other neurons are zero.*

Thus, DNF recovery solutions are networks where all terms in $f^*$ have corresponding neurons aligned with them. Furthermore, all other neurons are zero. In other words, a recovery solution encodes the DNF explicitly in the weights of the network (and thus the DNF can be easily recovered from the weights). The conditions on the $\lambda_i$ are required for the global optimality results to hold. We note that any DNF recovery solution perfectly classifies the data.

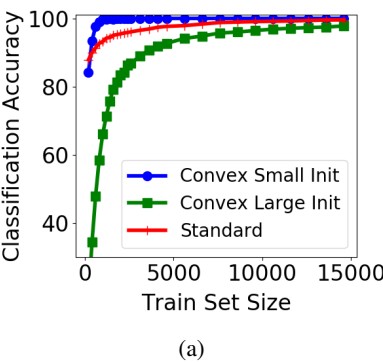 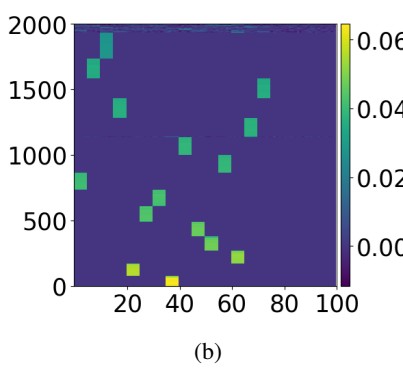 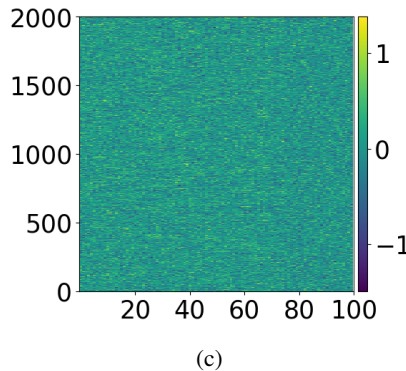

(a)           (b)           (c)

Figure 3: (a) **Accuracy Comparison for Different Models**: Test performance of a convex network with small Gaussian initialization, a convex network with large Gaussian initialization, and a standard two-layer networks with Xavier initialization. Here $D = 25$ and the target DNF has 4 terms of sizes 4, 5, 5 and 6. Each dot corresponds to the mean of 10 experiments with different initializations. (b) **SGD Learns a DNF recovery solution for** $D = 100$: The last 25 dimensions are noisy inputs whose corresponding literals do not appear in the DNF. [2] (c) **SGD With large initialization doesn't learn a DNF recovery solution:** The target function is the same as in Figure 3b. The only different is the initialization size. [2]

Next, we state our technical assumptions:

**Assumption 6.1.** *The output layer bias is fixed to $c = -1$, and $(\boldsymbol{W}, \boldsymbol{b})$ are learned.*

**Assumption 6.2.** $\mathbb{S}_{\boldsymbol{x}} = \mathcal{X}$, *i.e., we are in the population setting.*

Assumption 6.1 does not limit the expressive power of the network (see Theorem 3.1) and in the supplementary we show qualitatively that fixing $c = -1$ does not change the inductive bias of GF.[6]

Many works have studied the population setting (Assumption 6.2) as a proxy to understand the performance in the empirical case [Daniely and Malach, 2020, Brutzkus and Globerson, 2017]. The population setting is a good test-bed to understand the inductive bias of GF since the loss has multiple zero training error solutions in this case (e.g., memorization and DNF recovery solutions).

We now state the main result of this section.

**Theorem 6.1.** *Consider the minimum norm optimization problem Eq. (4) when learning a read-once DNF under Assumption 6.1 and Assumption 6.2. Then, any globally optimal solution $\boldsymbol{\theta}^* = (\boldsymbol{W}^*, \boldsymbol{b}^*)$ of Eq. (4) is a DNF-recovery solution.*

We next provide a high level sketch of the proof. The full proof is given in the supplementary.

First we notice two key properties of any perfect solution $\boldsymbol{\theta} = (\boldsymbol{W}, \boldsymbol{b})$ (Definition 3.1):[7]

---

[6]Surprisingly, without this assumption the theoretical analysis becomes substantially more difficult. We leave the extension to any $c$ for future work.

[7]Note that the set of perfect solutions is exactly the set of feasible solutions of Eq. (4).

(1) $\forall \boldsymbol{x} \in \mathbb{S}_+$ there exists $\mathbb{I} \subseteq [r]$ such that $\sum_{i \in \mathbb{I}} \boldsymbol{w}_i \cdot \boldsymbol{x} + b_i \geq 2$.

(2) $\forall \boldsymbol{x} \in \mathbb{S}_-, \forall i \in [r] \;\; \boldsymbol{w}_i \cdot \boldsymbol{x} + b_i \leq 0$.

These properties are together necessary and sufficient for solutions to be perfect. Next, we show the following upper bound on the bias of every neuron in a perfect solution, which depends on the weights of the neuron:

**Lemma 6.1.** *Under Assumption 6.1 and Assumption 6.2, if $\boldsymbol{\theta}$ is a perfect solution then every $i \in [r]$ satisfies:*

$$b_i \leq -\|\boldsymbol{w}_i\|_1 + 2 \sum_{n \in [K]} \max \left\{ \min_{j \in \mathbb{A}_n} \{w_{ij}\}, 0 \right\} \quad (6)$$

This upper bound turns out to be tight for optimal solution, as shown next. The following result characterizes the parameters of any globally optimal solution $\boldsymbol{\theta}^*$.

**Lemma 6.2.** *Under Assumption 6.1 and Assumption 6.2, if $\boldsymbol{\theta}^*$ is a globally optimal solution then:*

1. *For every $i \in [r]$:*

   1.1. *The bias achieves the upper bound in Lemma 6.1:*

   $$b_i^* = -\|\boldsymbol{w}_i^*\|_1 + 2 \sum_{n \in [K]} \max \left\{ \min_{j \in \mathbb{A}_n} \{w_{ij}^*\}, 0 \right\}$$

   1.2. $\boldsymbol{w}_i^* \geq 0$

   1.3. $\exists n \in [K]$ *such that neuron $i$ aligns with term $n$ or $\boldsymbol{w}_i^* = \boldsymbol{0}, b_i^* = 0$.*

2. *Neurons that align with the same term have the same alignment coefficient.*

3. *If $\mathbb{I} \subseteq [r]$ is the set of all neurons that align with term $n \in [K]$, then $\sum_{i \in \mathbb{S}} \lambda_i = 1$.*

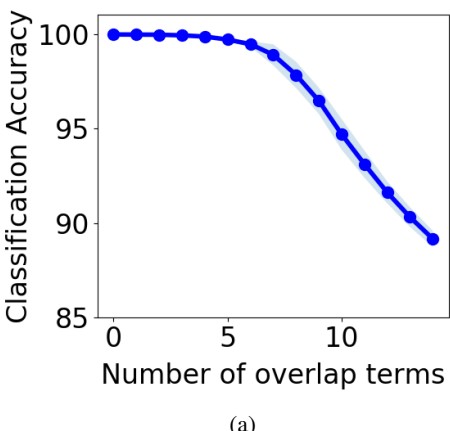
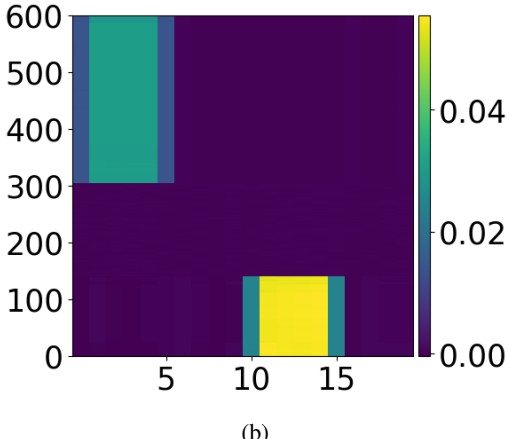

(a)            (b)

Figure 4: (a) **Evaluating the Effect of Overlap:** Here we experimented with $D = 100$ and all DNFs had 15 terms of size 5. We considered non read-once DNFs where the number of terms that share the variable $x_1$ varies. The $x$ axis shows the number of overlapping terms. The training size was $8,500$ for all DNFs. Each dot corresponds to the mean of 10 experiments with different initializations. (b) **SGD Does not Learn a DNF recovery solution:** We trained a convex network to learn the following 4-term DNF: $(x_1 \wedge x_2 \wedge x_3 \wedge x_4 \wedge x_5) \vee (x_2 \wedge x_3 \wedge x_4 \wedge x_5 \wedge x_6) \vee (x_{10} \wedge x_{11} \wedge x_{12} \wedge x_{13} \wedge x_{14}) \vee (x_{11} \wedge x_{12} \wedge x_{13} \wedge x_{14} \wedge x_{15})$ where $D = 20$. The training set size has 15,000 samples and the test classification accuracy is 100%. [2]

We prove the correctness of the properties one by one, where each proof relies on the correctness of the previous properties. The structure of the proof of all properties is similar: given a globally optimal solution, we assume by contradiction that it doesn't satisfy a specific property. Then we build a different perfect solution that satisfies this property and has a lower norm than the original globally optimal solution, thus contradicting optimality. The theorem directly follows from the aforementioned properties.

## 7   EMPIRICAL RESULTS

In this section we perform numerous experiments that support our analysis and show that our conclusions hold in different settings. For each experiment we show a sample of the empirical results due to space constraints. Further details and results are provided in the supplementary.

**Comparing convex and standard networks:** Our analysis focused on a convex network. Here we compare it to a standard two-layer network with trainable output layer. Figure 3a reports results, showing that the convex network outperforms the standard one for a DNF with $D = 25$. This shows that the convex network is a good model for studying inductive bias in our setting.

**Comparing large and small initializations:** In Figure 3a we compare a convex network with small initialization and a convex network with large initialization which is analogous to training in the NTK regime [Chizat et al., 2019]. The small initialization convex network performs better. We also show in Figure 3c that the small initialization network converges to a solution that aligns with the terms of the DNF

while the large initialization network does not.

**DNFs with large input dimension:** Here we show an experiment for learning a DNF with 15 terms of size 5 and $D = 100$. We learned the DNF using a convex network and SGD with small Gaussian initialization and 15,000 training samples drawn from the uniform distribution. In Figure 3b, we see that SGD converges to a solution that aligns with the terms of the DNF and has 100 % test accuracy.

**Non read-once DNFs:** Our theoretical work is restricted to read-once DNFs. To get a better understanding of what happens beyond the read-once case, we perform a series of experiments for learning DNFs with overlapping terms. Figure 4a shows that when we increase the number of overlapping terms, the generalization error gets worse.

Figure 4b shows an example of the neurons when learning a DNF with 4 overlapping terms. Here, the neurons do not align with the terms, and therefore the inductive bias is different from DNF recovery solutions.

The above results suggest that when the overlap is introduced to the learned DNFs, it becomes harder to recover the DNF and generalize well. This observation is in line with the fact that the known polynomial bound for learning monotone read-$k$ DNF [Mansour and Schain, 2001] increases with $k$. Indeed, $k$ is the number of times each variable can appear in the DNF and a larger value indicates that there is more overlap between the terms. Furthermore, known hardness results for learning general DNFs [Pitt and Valiant, 1988] also coincide with this empirical observation.

**Experiments on Tabular Datasets:** The fact that SGD recovers simple Boolean formulas is very attractive in the con-

text of interpretability. In Section 4 we showed that we can reconstruct DNFs under certain idealized assumptions (e.g., uniform distribution, read-once). However, our reconstruction method might produce meaningless reconstructions on datasets which are not uniform nor labeled with a read-once DNF. We tested our reconstruction method on three tabular UCI datasets kr-vs-kp, diabetes and Splice [Dua and Graff, 2017]. We note that these datasets do not contain personally identifiable information or offensive content.

Learning with our convex network resulted in test accuracies of 100%, 98% and 97% on these datasets, respectively. Our reconstruction method obtained a small DNF (6 terms of size less than 4) on kr-vs-kp with test accuracy 91%. For diabetes, the reconstruction method returned a large DNF (more than 10 terms) with test accuracy $81\%$. On Splice we got a 2-term DNF of sizes 2 and 3 with $95\%$ test accuracy. The latter is a very compact DNF with very small loss in accuracy, illustrating the potential of recovery on interpretability.

## 8  CONCLUSIONS

Understanding the inductive bias of neural networks for learning DNFs is an important and difficult theoretical challenge. In this work we focused on a setup of learning read-once DNFs under the uniform distribution with a convex network. We empirically observed that GF converges to solutions that align with the DNF terms. We then proved that GF cannot converge to solutions that memorize the training points, despite the fact that they minimize the training accuracy. We additionally proved that under certain assumptions, solutions that minimize the norm are solutions that align with the terms of the DNF. Together with recent results which show that GF is biased towards minimization of norms, this corroborates our empirical findings.

Our work has several limitations which are mainly due to the fact that analyzing nonlinear networks with nonlinear data is a major challenge:

1. We only consider a setup of uniform distribution and read-once DNF. We also restrict analysis to convex networks, but our empirical results actually suggest that convex networks may be preferable to standard ones in our setting.

2. We do not show an end-to-end convergence of GF to DNF recovery solutions.

3. We do not address the sample complexity of GF in our setting (however, intuitively, an inductive bias towards alignment will improve sample complexity, since alignment reduces the effective number of parameters the network uses).

Our work opens up many interesting directions for future work. For example, it would be interesting to understand if DNF recovery is possible for other distributions and DNFs that are not read-once. Another interesting direction is to understand the sample complexity of neural networks for learning DNFs and how it relates to DNF-recovery. Finally, it will be interesting to understand how learning dynamics in neural nets are related to other algorithms for learning DNFs.

## ACKNOWLEDGMENTS

This project was funded by the European Research Council (ERC) under the European Unions Horizon 2020 research and innovation programme (grant ERC HOLI 819080). AB is supported by a Google PhD fellowship.

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
