# OpenReview forum: "On the Inductive Bias of Neural Networks for Learning Read-once DNFs"
_auai.org/UAI/2022/Conference — UAI 2022 Poster_

### Official Review · Reviewer_wMsF · 2022-04-02

**Q2(1) Originality/Novelty:** 2
**Q2(2) Significance/Impact:** 2
**Q2(3) Correctness/Technical Quality:** 3
**Q2(6) Clarity Of Writing:** 1
**Q6 Overall Score:** 4
**Q8 Confidence In Your Score:** 3

**Q1 Summary And Contributions:**

A theoretical analysis of using a convex network to learn a positive read-once Disjunctive Normal Form, from a training set sampled from a uniform distribution. The main proofs are 1) the expressivity of the architecture, 2) gradient flow does not converge to memorizing solutions, and 3) under certain conditions the solutions minimizing the norm align with the DNF's terms. The empirical results illustrate the findings, and compare cases where not all assumptions hold (e.g. not read-once DNF).

**Q2 Assessment Of The Paper:**

More detailed information regarding each of these aspects is given below:

**Q2(4) Quality Of Experiments (Optional):**

3: Good: The experimental evaluation is adequate, and the results convincingly support the main claims.

**Q2(5) Reproducibility:**

3: Good: Key resources (e.g., proofs, code, data) are available and key details (e.g., proofs, experimental setup) are sufficiently well-described for competent researchers to confidently reproduce the main results.

**Q3 Main Strengths:**

+ The work seems thorough: it answers several questions and also (empirically) covers cases where the initial assumptions do not hold.
+ Novel
+ Paper is well structured
+ Reproducibility: provided code and mentioned experimental parameters (in the appendix).

**Q4 Main Weakness:**

- The problem formulation is not clearly written (cf. detailed comments).
- While the work appears thorough, details (of proofs and experiments) are often referred to the appendix which makes the paper a lot less self-contained and harder to read/verify.
- The theoretical contributions are mostly limited to read-once DNFs, which appears very constraining for applications.

**Q5 Detailed Comments To The Authors:**

## Comments

**The problem formulation is not clear.**
- Start with formally defining (read-once) DNF in logical terminology (literals, term, disjunction of terms) before describing the more numeric encoding of it. The current problem formulation is quite hard to read. Also write at least once in full all used abbreviations; 'disjunctive normal form (DNF)' does not occur in the text but it really should.
- Is $n$ in $x \sim \prod_{i=1}^n Bernoulli(0.5)$ supposed to be $D$?
- Mention $D$ = number of variables
- The task is to learn $f^*(x)$. Does that mean that the output must be a Boolean function in read-once DNF format, or does it mean that it must be a Boolean function (in any representational form) that can be represented as a read-once DNF (so the learned function $f^*(x)$ is not necessarily in a DNF format)?

While the overall work appears thorough, very often the main text refers to the appendix for details/proofs. The main text on its own becomes a lot less self-contained.

In case the population size is too large to check the actual accuracy it would be insightful to also report, on top of the sample based test accuracy, the model count of the actual DNF $f$, the learned DNF $f'$, and $f \vee f'$.

I did not verify the proofs in the appendix, and found the proofs within the paper not always easy to follow. In particular for the proof of Theorem 6.1 I would have liked to see more details within the main paper itself.



## Questions

Q1) Figure 1's axes are not labelled. What are they? I assumed x-axis is the hidden units and y-axis is all potential inputs but there should be $2^9$ of those instead of $600$, so I'm confused.

Q2) "nonlinear read-once DNF" was never defined. What is a *nonlinear* read-once DNF, in logical terminology (literals, terms, ...)?

Q3) Sect 7. "The fact that SGD recovers simple Boolean formulas is very attractive in the context of interpretability" - but the previous paragraph empirically showed that when learning from data sourced from a DNF with overlapping terms, the solution is not a DNF recovery solution? Those would not be easily interpretable?

Q4) Sect 7. "Non read-once DNFs" - are the literals still all positive or both negatives and positives?

Q5) "To show each property we assume by contradiction that it holds and construct a perfect solution with lower norm. This leads to a contradiction since the solution has minimal norm." - "To show **that** each property **holds**"? The proof is not very clear to me, how does it rely on the properties?

Q6) The focus is on learning a read-once DNF from training data (problem formulation). If the learned function did not have to be a read-once DNF, then the problem reads like a binary classification problem for which methods exist (e.g. a learned decision tree can easily be turned in a DNF)? What makes the read-once DNF constraint specifically interesting? Are those common/a good approximation in practice? I understand that from a theoretical perspective it could be interesting to initially restrict the input to read-once DNFs, to make the analysis easier, but why constrain the output to a read-once DNF too (do-we? See comment later on problem formulation)?

Q7) Which proofs/theoretical statements rely on $\mathcal{D}$ being uniform?

Q8) Which proofs/theoretical statements rely on the read-once restriction?

Q9) In Figure 2a, only with Training set size $> 2000$ does the SQ algorithm seem to more consistently achieve 100% accuracy. However, I had the impression that if SQ was given the entire population ($2^9$ samples here) it would be 100% accurate. Is this not true or did the training set just not happen to contain the entire population?

Q10) In Figure 2a: did each run only differ in initialization, or also a different training and test set? Did the 3 approaches learn from the same training set and evaluated on the same test set?


## Textual remarks:

* Fig 3b (and similar) are missing axis labels

* "Learning DNFs is hard [Pitt and Valiant, 1998]" -> "Learning read-once DNFs is hard", to be more specific

* 'the a set' -> 'the set'

* "We perform experiments on DNFs of higher dimension" - 'higher dimension' is not clear, does this mean more terms or more variables or ? Should this be 'read-once' DNF?

* Fig 4 caption, "The training size was 8, 500 for all DNFs." - is this 8500 (without space)?

* "Their results suggest that the inductive bias of GF is to KKT points or global solutions of minimum norm problems" and later "In our theoretical analysis, we apply the results of Lyu and Li [2020], Ji and Telgarsky [2020], which show that GF is biased towards KKT points of min-norm problems." - Did they proof (~show) this or 'suggest' this?

* In Assumption 5.1, what is $n$?


**Q7 Justification For Your Score:**

Although I am not totally convinced by the read-once restriction (its relevance on applications), I consider this a light remark as the authors analyse an interesting problem, and appear to do so thoroughly. I however found the paper quite hard to follow. I suggest to make the problem formulation more precise, and make the paper more self contained by moving more details (particularly of proofs) into the paper itself. It can be written more concisely as to fit more of the information.

**Q9 Complying With Reviewing Instructions:**

1: Yes.

---

### Official Review · Reviewer_Txfk · 2022-04-12

**Q2(1) Originality/Novelty:** 3
**Q2(2) Significance/Impact:** 1
**Q2(3) Correctness/Technical Quality:** 3
**Q2(6) Clarity Of Writing:** 3
**Q6 Overall Score:** 5
**Q8 Confidence In Your Score:** 4

**Q1 Summary And Contributions:**

The paper investigates the ability of two-layer neural networks to learn read-once DNFs. That is, negation-free formulae in disjunctive normal form that use each variable only once.

**Q2 Assessment Of The Paper:**

More detailed information regarding each of these aspects is given below:

**Q2(4) Quality Of Experiments (Optional):**

2: Fair: The experimental evaluation is weak: important baselines are missing, or the results do not adequately support the main claims.

**Q2(5) Reproducibility:**

3: Good: Key resources (e.g., proofs, code, data) are available and key details (e.g., proofs, experimental setup) are sufficiently well-described for competent researchers to confidently reproduce the main results.

**Q3 Main Strengths:**

* There is a good theoretical analysis of the behavior of neural networks on this type of problem, and experiments that align with this theory.

* There are also experiments that show that the 'read-once' restriction is needed, which means that the results are 'tight', and it gives an interesting example of an unlearnable function.

**Q4 Main Weakness:**

* It would be helpful to define "DNF". I think the authors are talking about boolean functions in disjunctive normal form, but they never explicitly say.

* The problem of learning read-once DNFs seems to be mostly theoretical.

* The experiments are limited to a a few fixed DNFs.

* The experiments are not described in enough detail. What learning method is used? What parameters? What are the lines in figure 2a?

* Notation is not introduced properly

* Some of the related work seems only distantly related

* The theorems are obvious/not very surprising
  * Theorem 3.1 looks like a variant of the universal approximation theorem,
  * Theorem 5.1 is an obvious consequence from gradient flow being a norm minimizing solution, as mentioned in section 3


**Q5 Detailed Comments To The Authors:**

Notation is not introduced properly:
 * [D], which seems to mean the set {1,2,..,D}
 * "x ∼ \prod Bernoulli(0.5)"
    but x has entries in {-1,+1}, Bernoulli usually implies {0,1}

Figure 1: how were these networks constructed?

There are no weights for second network layer in (1). Is that too limiting?

Figure 3: what is "small Gaussian initialization"?

Definition 6.2 "and all other neurons are zero."
 What does that mean?

Theorem 6.1: is read-once an implicit assumption here? It should be explicit in the statement of the theorem.

The experiment in figure 4a is the most interesting part of the paper to me, showing a big contrast between the ability of a neural network to learn general DNFs vs read-once DNFs.

**Q7 Justification For Your Score:**

The paper is correct, and introduces some interesting ideas. But the problem of read-once DNFs is very artificial, and as a result the impact of the paper is probably small.
The experiments are too limited, and not described in enough detail.

**Q9 Complying With Reviewing Instructions:**

1: Yes.

---

### Official Review · Reviewer_reoo · 2022-04-13

**Q2(1) Originality/Novelty:** 3
**Q2(2) Significance/Impact:** 3
**Q2(3) Correctness/Technical Quality:** 3
**Q2(6) Clarity Of Writing:** 4
**Q6 Overall Score:** 7
**Q8 Confidence In Your Score:** 3

**Q1 Summary And Contributions:**

This paper studies the inductive bias for learning read-once DNFs by ReLU networks with one hidden later and a fixed output layer. Read-once DNFs are Boolean formulas in disjunctive normal form where every logical variable appears once (the authors also seem to only consider monotone DNFs, if I understood correctly the setting).

**Q2 Assessment Of The Paper:**

More detailed information regarding each of these aspects is given below:

**Q2(4) Quality Of Experiments (Optional):**

3: Good: The experimental evaluation is adequate, and the results convincingly support the main claims.

**Q2(5) Reproducibility:**

4: Excellent: Key resources (e.g., proofs, code, data) are available and key details (e.g., proof sketches, experimental setup) are comprehensively described for competent researchers to confidently and easily reproduce the main results.

**Q3 Main Strengths:**

The authors contribution rests both on experimental evaluation and on new theoretical results. Experimentally, the authors show that the studied networks converge to solutions which generalize well (in particular better than two-layer MLP and another method for learning DNFs). Moreover, the trained networks have neurons aligned with terms of the DNFs. Theoretically, the authors show, under certain assumptions that (i) gradient flow does not learn solutions that “memorize” the data, and (ii) that the learned network learns to reconstruct the DNF.

Interesting question: The paper studies an interesting question, which is the inductive bias for learning read-once DNFs. I really appreciate not only the results but also the problem and its specification.

Nice mix of theory and experiments: The theory and experiments nicely complement each other. It is nice that the authors also study settings that go beyond their theoretical analysis (DNFs that violate the read-once property).


**Q4 Main Weakness:**

Especially, for the second result, however, there are stronger assumptions used. First, the sample is assumed to be the whole set X (i.e. all possible instances) and (ii) it is assumed that the learned solution will be a minimum-norm solution (the authors give compelling evidence from the literature but a proof does not exist—it may likely be very difficult, so this is not a criticism. The assumption on the “population setting” is quite strong.

However, I think that such assumptions might be necessary for a work like this, so I do not consider these to be big weaknesses.

**Q5 Detailed Comments To The Authors:**

Overall, the paper is well written.

I was just wondering what happens if the DNF contains just one term that contains all propositional variables. Will the solution not be memorizing? If not then why not?

**Q7 Justification For Your Score:**

The paper studies an interesting question. The results are interesting even though they depend on quite strong assumptions. The paper reads nicely.

**Q9 Complying With Reviewing Instructions:**

1: Yes.

---

### Official Review · Reviewer_qs5A · 2022-04-13

**Q2(1) Originality/Novelty:** 3
**Q2(2) Significance/Impact:** 3
**Q2(3) Correctness/Technical Quality:** 3
**Q2(6) Clarity Of Writing:** 4
**Q6 Overall Score:** 8
**Q8 Confidence In Your Score:** 3

**Q1 Summary And Contributions:**

The paper demonstrates strong theoretical and empirical connections between the inductive bias of gradient flow applied to neural networks with one hidden layer and terms in DNF formulas, following recent work that connects GF to KKT points. This is an important step towards understanding the inductive bias of modern neural networks.

**Q2 Assessment Of The Paper:**

More detailed information regarding each of these aspects is given below:

**Q2(4) Quality Of Experiments (Optional):**

3: Good: The experimental evaluation is adequate, and the results convincingly support the main claims.

**Q2(5) Reproducibility:**

3: Good: Key resources (e.g., proofs, code, data) are available and key details (e.g., proofs, experimental setup) are sufficiently well-described for competent researchers to confidently reproduce the main results.

**Q3 Main Strengths:**

1. Important step in understanding the inductive bias of modern neural networks.
2. Convincing theoretical and empirical analysis.
3. Excellent presentation.

**Q4 Main Weakness:**

1. Only one hidden layer is considered, but this is sufficient to represent read-once DNF.
2. Only uniform distributions.
3. No sample complexity results.

**Q5 Detailed Comments To The Authors:**

I really enjoyed reading your paper and as a non-expert got a lot out of it. It is very well-presented and I have only a few minor comments.

I don't think the abbreviation KKT is explained anywhere in the paper.

At the bottom right of p.3 the citation "Amos et al. [2017]" should be [Amos et al. 2017].

The paper makes quite specific contributions and I think the title doesn't adequately reflect this.

**Q7 Justification For Your Score:**

I am not an expert in the area, but I found the approach and account very compelling. The only way in which I could see this paper being rejected is if there are technical issues or limited novelty, which I hope the other reviewers will be better able to judge.

**Q9 Complying With Reviewing Instructions:**

1: Yes.

---

### Decision · Program_Chairs · 2022-05-15

**Decision:**

Accept (Poster)

**Comment:**

Meta Review: Most reviewers appreciated the insights provided on learning neural networks.  Some reviewers also had some concerns about readability; hopefully they provided enough feedback to improve the paper.